# Roles of Topoisomerases in Heterochromatin, Aging, and Diseases

**DOI:** 10.3390/genes10110884

**Published:** 2019-11-01

**Authors:** Seung Kyu Lee, Weidong Wang

**Affiliations:** Lab of Genetics and Genomics, National Institute on Aging, National Institute of Health, Baltimore, MD 21224, USA; seungkyu.lee@nih.gov

**Keywords:** topoisomerase, Top3β, Top2, heterochromatin, transposon, aging, disease

## Abstract

Heterochromatin is a transcriptionally repressive chromatin architecture that has a low abundance of genes but an enrichment of transposons. Defects in heterochromatin can cause the de-repression of genes and transposons, leading to deleterious physiological changes such as aging, cancer, and neurological disorders. While the roles of topoisomerases in many DNA-based processes have been investigated and reviewed, their roles in heterochromatin formation and function are only beginning to be understood. In this review, we discuss recent findings on how topoisomerases can promote heterochromatin organization and impact the transcription of genes and transposons. We will focus on two topoisomerases: Top2α, which catenates and decatenates double-stranded DNA, and Top3β, which can change the topology of not only DNA, but also RNA. Both enzymes are required for normal heterochromatin formation and function, as the inactivation of either protein by genetic mutations or chemical inhibitors can result in defective heterochromatin formation and the de-silencing of transposons. These defects may contribute to the shortened lifespan and neurological disorders observed in individuals carrying mutations of Top3β. We propose that topological stress may be generated in both DNA and RNA during heterochromatin formation and function, which depend on multiple topoisomerases to resolve.

## 1. Introduction

Topoisomerases are essential enzymes that can resolve topological problems generated during DNA and RNA metabolism. They have a unique ability to catalyze transient cleavage and passage of single or double-strand DNA, and then, reseal the broken ends after the passage. This ability allows them to relax supercoiled DNA and decatenate inter-tangled DNA produced during transcription, replication, recombination, and chromosome condensation and segregation [1]. Topoisomerases can be categorized into two main types: Type I and II, which can cleave and re-ligate single and double stranded DNA, respectively. Each type of topoisomerases can be further divided into A or B depending on the mechanism of the strand passage. Mammals have two Type IA topoisomerases, Top3α, Top3β; two Type IB topoisomerases, Top1 in nucleus and Top1mt in mitochondria; and two Type IIA topoisomerases: Top2α and Top2β. Type IB topoisomerases can cleave one strand of DNA, allow one broken strand to rotate round the unbroken strand to remove all supercoils, and then re-ligate the broken ends. Type IIA and IIB can catalyze ATP-dependent dsDNA passage reactions, enabling catenation and decatenation of DNA. 

Topoisomerases are essential for viability of many organisms. For example, Top1 is essential in multicellular organisms, whereas Top2 is essential in all organisms. Specifically, mouse and *Drosophila* lacking either Top1 or Top2 (α and β) exhibits lethality in early developmental stage [2,3,4]. While Top3β is not an essential gene in these species, mouse lacking Top3β displays shortened lifespan, higher incidence of aneuploidy in germ cells, increased autoimmunity [5,6,7], and abnormal synapse formation [8]. Importantly, topoisomerase mutations have been associated with various human diseases, such as cancer, neurodegeneration, autism, and autoimmune diseases [9,10,11,12]. Moreover, drugs targeting topoisomerases have been widely used in treatment of cancers [13]. Thus, understanding the mechanism of these enzymes may allow for development of drugs in treatment of more human disorders. 

In eukaryotes, DNA wraps around histone octamers to form nucleosomes, which are then assembled into higher-order chromatin structures. Formation of the chromatin allows efficient storage of genetic information, but also restricts the access of DNA to machinery of transcription, replication, and other processes. Topoisomerases often play a positive role to make the chromatin DNA accessible for these processes. For example, during transcription initiation, topoisomerases can facilitate disassembly of the nucleosome-DNA structures and reduce histone density to enable transcription machinery to access promoters [14,15]. In addition, Top1 and Top2 can regulate histone modifications to generate a conducive chromatin environment for binding of transcription factors and Pol II to promoters [16,17]. Moreover, Top1 and Top2α have been shown to work with chromatin remodeling enzymes during transcription and genome stabilization [14,18,19].

Chromatin can be classified into two main domains: euchromatin, which is transcriptionally active and enriched with genes; and heterochromatin, which is transcriptionally inactive, has low abundance of genes but high abundance of transposons [20,21]. The distinction between heterochromatin and euchromatin can be characterized by different histone modifications. Euchromatin exhibits active histone marks including acetylation of H3K36 and H3K4 methylation, while heterochromatin is associated with repressive histone marks such as heavily methylated H3K9 or global hypoacetylation [22]. Upon modification of histones, key proteins for heterochromatin formation, such as Heterochromatin Protein 1 (HP1) or Histone methyltransferase (HMT), are recruited and a compact chromatin structure is formed. There are two types of heterochromatin: facultative and constitutive heterochromatin. Both types are transcriptionally repressed and exhibit high nucleosome density. Constitutive heterochromatin is generally formed at pericentric region, which consists of repetitive tandem repeats with more static structure. Facultative heterochromatin can be temporally and spatially decondensed and condensed within euchromatin region. Contrary to the constitutive heterochromatin that is generally marked by H3K9me3, facultative heterochromatin often associates with Polycomb group protein mediated H3K27me3 [23].

There have been many reviews discussing the roles of topoisomerases in transcription of genes from euchromatin [12,24,25,26]. In this review, we discuss the roles of topoisomerases in heterochromatin and their importance in aging and various diseases.

## 2. Importance of Heterochromatin

### 2.1. Heterochromatin Is Critical for Transcriptional Silencing of Transposons

A main function of heterochromatin is protecting the underlying genome from being accessed and transcribed by transcriptional machinery. This is important because the major components of heterochromatin are transposable elements (TEs) and tandem repetitive sequences. TEs are mobile genetic elements that can jump and randomly insert within a genome. They are present in all living organisms, and are major constituents of eukaryotic genome (more than 40% of human genome, 37% of mouse genome, and more than 80% of maize genome) [27,28,29]. TE and repeat integration or excision can generate chromosomal reorganization and other genetic alterations [30]. They can produce deleterious effects by landing in essential genes, causing mutations or mis-regulation. Consequently, TEs have been shown to be a potent mutagen that can cause genomic instability, aging, and various diseases, including cancers and neurological disorders [31,32,33]. To suppress their harmful effects, TEs and repeats are subjected to transcriptional and post-transcriptional silencing mechanisms [34,35,36]. Heterochromatin has a crucial role in suppressing TE and repeat expression through diverse mechanisms.

### 2.2. Loss of Heterochromatin May Be a Cause of Aging and Premature Aging Syndromes

Aging is one of the many well-studied physiological processes that has been associated with heterochromatin loss. In the ‘heterochromatin loss model of aging’ [37], it was proposed that heterochromatin is established in the early developmental stage and maintained through life. This epigenetic heterochromatin structure must be regenerated each time DNA is replicated or repaired. DNA damage and cell division may be the major perturbing factors triggering heterochromatin loss. This could result in alteration of age-related gene expression at the periphery of the heterochromatin domains. Indeed, many studies have shown that increasing heterochromatin formation can extend lifespan. For example, over-expression of heterochromatin protein HP1 in *Drosophila* increases the longevity [38]. In human, the loss of heterochromatin underlies two well-characterized progeroid syndromes: Werner syndrome (WS) and Hutchinson-Gilford Progeria Syndrome (HGPS) [39,40,41,42], Notably, the HGPS model exhibits up-regulation of satellite III repeats, suggesting that de-silencing of repeats in loss of heterochromatin may contribute to aging [40].

### 2.3. Heterochromatin Loss May Increase Cancer Risks

Loss of heterochromatin has been shown to impair chromosome segregation, nucleosome compaction, and DNA repair in multiple cancer cells [43,44]. In several cancer progression models, loss of HP1 or H3K9 methylation in heterochromatin has been shown to correlate with cancer progression and tumorigenesis [43,45,46]. One potential mechanism is that the reduction in heterochromatin may trigger mutagenic retro-transposition of satellite DNA and transposons. Transposition of these mobile elements may result in inactivation of tumor suppressor genes or activation of oncogenes. For example, highly active retro-transposition of a mammalian retrotransposon LINE-1 has been identified in multiple types of cancers, suggesting that heterochromatin mediated silencing of LINE-1 is disrupted [47]. It has been proposed that the insertion of LINE-1 in specific loci can disrupt tumor suppressors or oncogenes, resulting in tumorigenesis [48].

### 2.4. Heterochromatin and Neurological Disorders

Alteration in heterochromatin has been linked to several neurological disorders, including neurodegeneration-associated tauopathy (such as Alzheimer’s disease). Dysfunction of the hyperphosphorylated Tau protein is one of the main causes of tauopathy [49]. *Drosophila* tauopathy models, which transgenically express human *tau^R406W^* mutant protein, display neurodegenerative phenotype along with loss of heterochromatin, as shown by reduced H3K9 methylation and HP1a [50]. Moreover, disrupting heterochromatin structure enhances the neurodegeneration phenotype of flies that over-express tau, whereas enhancing heterochromatin formation reduces neurodegeneration, suggesting that heterochromatin loss has an important role in tau-induced neurodegeneration [50]. Indeed, several studies reported that brains of human Alzheimer’s disease patients exhibit loss of heterochromatin and de-silencing of genes [50,51]. Two other studies also show that a significant alteration of transposon expression in human and fly tauopathy brains, implicating that dysregulation of transposons in heterochromatin may cause neurotoxicity [52,53].

## 3. Multiple Topoisomerases Function in Heterochromatin

Heterochromatin is a stable but dynamic structure. Its formation and maintenance require numerous factors including: histone modification enzymes, chromatin assembly factors, heterochromatin associated structural proteins, chromatin remodeling enzymes [54,55], and several topoisomerases, as described below. 

### 3.1. Top1

Top1 has been shown to be required for heterochromatin structure and histone modifications. An ultrastructural study showed that Top1 inhibitors, Camptothecin (CPT) and Rebeccamycin, can cause unpacking of heterochromatin in *Trypanosoma cruzi* [56]. Similarly, CPT treatment of human HCT116 cells induces decompression of heterochromatin with altered histone modifications [57]. There is also evidence that Top1 inactivation may disrupt a main function of heterochromatin, transcriptional silencing of transposons [58]. The mechanism of how Top1 inhibition leads to disruption of heterochromatin remains unclear. One study shows that depletion of Top1 in human HEK293 cells results in excessive RNA-DNA hybrid (R-loop) formation in heterochromatin domains, implicating a role of Top1 in regulating R-loop homeostasis in heterochromatin [59]. The R-loop is formed by nascent RNA entangled with template DNA during transcription; and it has been shown as a mediating structure during RNA-induced heterochromatin formation [60]. It is possible that Top1 inhibition may alter heterochromatin formation by disrupting R-loop homeostasis.

### 3.2. Top2

A fraction of Top2 has been detected in heterochromatin in *Drosophila* [61,62]. Subsequent studies revealed that Top2 is required for heterochromatin condensation, as *Drosophila Top2* mutant cells have decondensed heterochromatin with increased aneuploidy and polyploidy [63]. In addition, Top2 depletion in *Drosophila* ovary results in poorly segregated heterochromatin, whereas euchromatin separation is normal during meiosis [64], suggesting that Top2 is specifically needed for segregation of heterochromatin region, which is resolved during cell cycle progression. Moreover, Top2 is needed for transcriptional silencing in heterochromatin, as displacement of Top2 from Satellite III in heterochromatin or chemical inhibition of Top2 activity can disrupt heterochromatic silencing of a reporter gene [65].

In studies from mammalian cell lines, Top2 is found to be required for chromosome condensation, segregation; and assembly of heterochromatin [66]. Silencing or chemical inhibition of Top2 induce chromatin reorganization and heterochromatin changes in mouse and human cell lines [67]. Moreover, Top2 inhibition by an inhibitor, etoposide, results in increased expression and mobility of repeats and transposons [68,69], consistent with the notion that Top2, like Top1, is also needed for heterochromatic silencing of transposons. 

How does Top2 function in heterochromatin? Recent findings from the Crabtree lab suggest that Top2 can work with chromatin remodeling complexes, BAF, to make facultative heterochromatin more accessible to transcription factors [70] (Figure 1). The mammalian BAF complexes are ATP-dependent chromatin remodeling complexes that can alter chromatin structure to make the DNA more accessible to machinery of transcription, replication, repair, and other processes on DNA [71,72,73]. In an early study by Crabtree’s group, they found that BAF complexes can directly interact with Top2α to facilitates its access to approximately 12,000 sites across the genome [19]. They demonstrated that cells mutated in the BAF complex components are defective in decatenation of newly replicated sister chromatids, which is a phenotype of cells inhibited of Top2α activity. The findings that BAF complex is needed by Top2α to prevent DNA entanglement during mitosis provides an explanation for the observation that BAF subunits are mutated in a large fraction of human cancers [72,73]. In a later study, Crabtree’s group used an *in vivo* Chromatin indicator Assay (CiA), which can induce chromatin remodeling by recruiting a chromatin regulator to a designed locus upon Chemical Inducer of Proximity. They demonstrate that Top2α is required for an early process during BAF-mediated chromatin remodeling of facultative heterochromatin. They also found that this process is specifically dependent on Top2α, but not Top1, suggesting that the specific topological state of facultative heterochromatin is catenated, which can be resolved by Top2α; rather than supercoiled, which can be resolved by Top1. Their findings imply that decatenating DNA within facultative heterochromatin by Top2α can make the chromatin more accessible for chromatin remodelers as well as transcription factors [70] (Figure 1). The data also suggest that facultative heterochromatin and accessible chromatin likely have different DNA topologies, with the former being more catenated than the latter.

In the same study, Crabtree and colleagues discovered that Top2α has another function: it can facilitate reformation of facultative heterochromatin from accessible chromatin [70]. They found that this function of Top2α does not depend on BAF complex. Their findings suggest that Top2α may promote catenation of DNA during heterochromatin formation (Figure 2).

### 3.3. Top3β

#### 3.3.1. Top3β Is a Dual-Activity Topoisomerase

Our group has recently discovered that Top3β is required for heterochromatin formation [74], as are other topoisomerases mentioned above. However, its mechanism appears to be different from others. This may not be a surprise because Top3β is the only dual activity topoisomerase in animals, and can bind and change topology of both DNA and RNA. Top3β is conserved from invertebrate to mammals, and is required for normal aging and genome stability in mice, as evidenced by findings that *Top3β* mutant mice exhibit reduced lifespan and fertility, chromosomal abnormality, and increased cell death [5,6,7]. Top3β contains an RNA-binding domain, possesses RNA topoisomerase activity *in vitro*, and is the major mRNA-binding topoisomerase in cells [8,75,76]. It should be mentioned that the RNA topoisomerase activity is conserved in type IA topoisomerases from all three domains of life: bacteria, archaea, and eukarya [75,76]; and most, but not all, type IA topoisomerases carry dual topoisomerase activity [76]. Therefore, Top3β and other type IA topoisomerases may act not only in processes of DNA, but also RNA. 

#### 3.3.2. Top3β Is Required for Heterochromatin Formation and Silencing of Transposons

In 2018, our group reported that Top3β is required for heterochromatin formation and transcriptional silencing using *Drosophila* model system. We demonstrate that Top3β, along with its partner TDRD3, biochemically and genetically interact with the RNA-induced silencing complex (RISC), which consists of RNA helicase p68; RNAi effector Argonaut 2 (AGO2); and two RNA-binding proteins (RBPs), FMRP, and VIG; in this process [74] (Figure 3A,B). This interaction is likely important because the RISC complex has previously been shown to promote heterochromatin formation and silencing of transposons in *Drosophila* [77,78,79,80]. 

Indeed, *Top3β* mutant flies resemble those of RISC mutants in several regards, including suppression of position effect variegation (PEV), reduction of HP1 and H3K9 methylation in heterochromatin, and de-silencing of transposons. Moreover, *Top3β* exhibits a strong genetic interaction with siRNA biogenesis enzyme, *Dcr-2;* and RISC components: RNA helicase *RM62* and RNAi effector, *AGO2*. These data suggest that Top3β may work with RISC to promote heterochromatic silencing, possibly by recruiting HP1 and other heterochromatin components. The fact that Top3β interacts with an RNA processing complex, RISC, suggests that it may solve RNA topological problems during heterochromatin function. Consistent with this notion, we showed that both RNA binding and catalytic activity of Top3β are required for Top3β-mediated heterochromatin formation and TE silencing. These findings provide a new mechanism for how topoisomerases may work: not only can they promote transcription activation by resolving supercoiled DNA, but also transcriptional repression by facilitating heterochromatin formation. 

Several key questions remain unanswered regarding the mechanism of Top3β in heterochromatin formation. One question is whether the Top3β-TDRD3 complex acts on DNA or RNA. The current evidence implies that it may act on both. For example, ChIP-seq experiments revealed binding of Top3β to a small number of DNA loci within heterochromatin domains, and these sites correlate with those of a RISC component, AGO2. Some of these sites also overlap with those of HP1. These findings support a previously proposed mechanism of “nucleation and spreading” [35,81]: RISC and Top3β may bind a small number of loci to nucleate the initial assembly of heterochromatin, which may then recruit additional components to spread to other regions. 

Conversely, there is also evidence that Top3β may act on RNA. In particular, Top3β can physically and functionally interact with the RNA processing complex, which includes an RNA helicase, p68. The interaction between a Type1A topoisomerase and a helicase has been observed for Top3α (which is the paralogue of Top3β) and BLM DNA helicase. It was shown that Top3α and BLM cooperate to resolve complex DNA structures, such as double-Holliday junctions [82,83]. We therefore postulated a potential mechanism of Top3β on RNA, based on the mechanism of Top3α and BLM. Top3β may work with p68 to make the nascent RNAs transcribed from heterochromatin more accessible to siRNA-guided RISC complex (Figure 3C). Possibly, p68 helicase may unwind the secondary and other higher order structures in the nascent RNAs, which may produce supercoiled or tangled RNA structures. Top3β may resolve these structures, allowing nascent RNA to be accessible to base-pairing interactions during siRNA-guided heterochromatin assembly.

It should be mentioned that mammalian Top3β-TDRD3 complex differs from its *Drosophila* counterpart in that it lacks stable association with AGO2, as evidenced by the absence of AGO2 in Top3β or TDRD3 immunoprecipitation from human cancer cell lines [8]. This result is consistent with the findings that mammalian TDRD3 lacks a recognizable AGO2-binding domain, which was detected in *Drosophila* TDRD3 [74]. These data argue that the mechanism by which Top3β-TDRD3 complex works with RISC complex could be different in mammals versus flies. Notably, we and others have consistently observed that TDRD3 interacts with a RISC component, FMRP [8,84,85]. FMRP has been reported to interact with AGO2 [86], and to regulate AGO2 binding to its target mRNAs in human cancer cell lines [87,88,89]. Thus, one possibility is that mammalian Top3β-TDRD3 may interact indirectly with AGO2, and this interaction is mediated by FMRP. It remains to be determined if Top3β-TDRD3 functionally works with AGO2 in promoting heterochromatin formation and silencing of transposons in mammals. 

## 4. Importance of Topoisomerase Function in Heterochromatin

Because topoisomerases can participate in a variety of processes on DNA and RNA, one question is whether their function in heterochromatin is relevant. In the case of Top3β, its function in heterochromatin could be very important, because it may account for several deleterious phenotypes observed in humans and mice with *Top3β* mutations. For example, *Top3β*-KO mice display reduced lifespan [6]. It is known that loss of heterochromatin and de-silencing of transposons could be a driver for aging [20,90]. Thus, the impaired function of Top3β in heterochromatin and transposon silencing may contribute to the aging phenotype in the knockout mice. In human, *Top3β* mutations have been linked to neurological and mental disorders, including: schizophrenia, autism, epilepsy, intellectual disability, and cognitive impairment [11,91,92,93,94], suggesting that *Top3β* mutations may cause neurological defects. Similarly, mouse and *Drosophila* with *Top3β* mutations also show abnormal synapse formation [8]. Heterochromatin and silencing of TEs have been found to be important in neurodevelopment. Dysregulation of heterochromatin can impair normal gene expression during neuronal development, leading to neurological diseases in humans [95]. Moreover, activation of TEs has been observed is in a variety of age-associated neurogenerative diseases [96,97,98], including Alzheimer’s disease [52,53]. It is therefore possible that the defective function of Top3β in heterochromatin may contribute to the neurological and mental disorders. 

It should be mentioned that *de novo* mutation in BAF complexes has been linked to neurological disorders, including autism and schizophrenia [99,100,101,102]. Because BAF complexes can cooperate with Top2α to promote access of facultative heterochromatin to transcription, it is possible that defective heterochromatin function may contribute to these diseases in patients with BAF mutations. 

## 5. Conclusive Remarks

Increasing evidence shows that defective heterochromatin and dysregulation of TEs may be a cause for aging, cancer, and neurological disorders. Several topoisomerases, Top1, Top2, and Top3β, are all important for heterochromatin formation and silencing of transposons. These topoisomerases have distinct properties and can solve different topological problems—Top1 can relax DNA supercoils by swiveling; Top2 can catenate or decatenate dsDNA; and Top3β can catalyze strand passage reactions for both DNA and RNA. These data suggest that there may exist different topological problems in heterochromatin, which require multiple topoisomerases to resolve. The heterochromatin function of Top3β could be particularly important because its mutations in mouse can cause shortened life, and its mutations in human have been linked to neurological disorders. 

There are many unanswered questions regarding mechanisms of topoisomerases. Are there both DNA and RNA topological problems during heterochromatin formation? If so, what types of structures? Entangled dsDNA, RNA/DNA hybrid, entangled RNA, or supercoiled DNA or RNA? It seems that all topoisomerases can work with ATP-dependent remodelers (BAF, FACT, and p68 helicase). It would be important to develop *in vitro* and *in vivo* systems to investigate the underlying mechanism. Do different topoisomerases act in common or different pathways to promote heterochromatin formation and silencing of transposons? Genetic studies in model systems or cell lines can be used to address this issue. 

## Figures and Tables

**Figure 1 genes-10-00884-f001:**
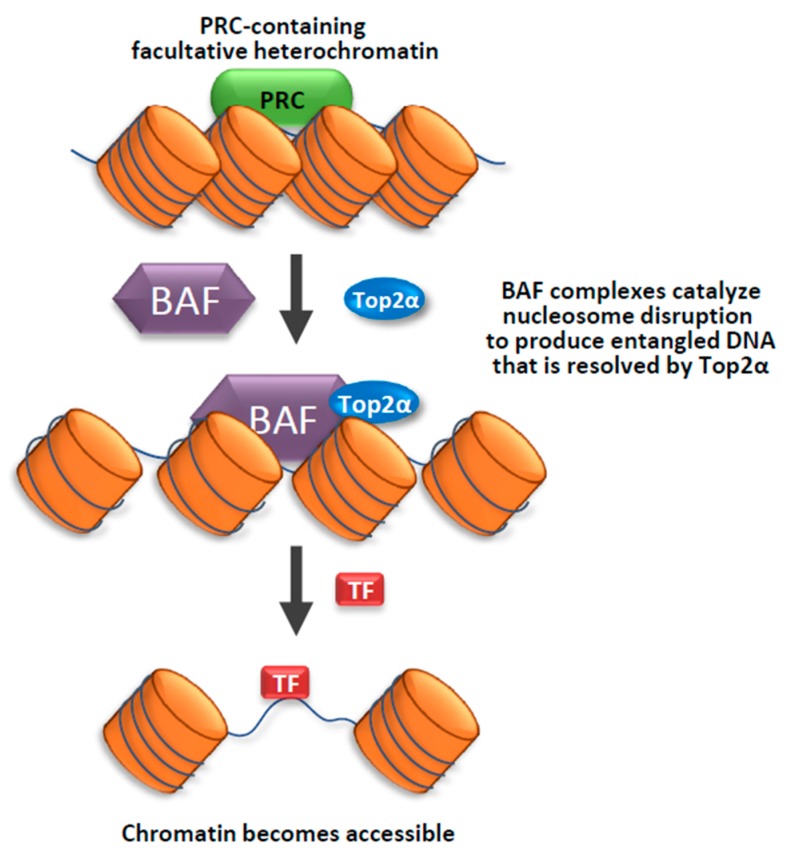
Top2α and the BAF complex cooperate to facilitate access to facultative heterochromatin. A model illustrating how Top2α and the BAF complex may work together to facilitate access to facultative heterochromatin. The BAF complex uses its ATP-dependent chromatin remodeling activity to disrupt the Polycomb group (PcG) complex containing facultative heterochromatin. Top2α is recruited in the early stage of chromatin remodeling to resolve the topological stress produced by the BAF complex, possibly using its DNA decatenation activity. This model is based on findings in the paper of Miller et al. (Nat. Struct. Mol. Biol., 2017).

**Figure 2 genes-10-00884-f002:**
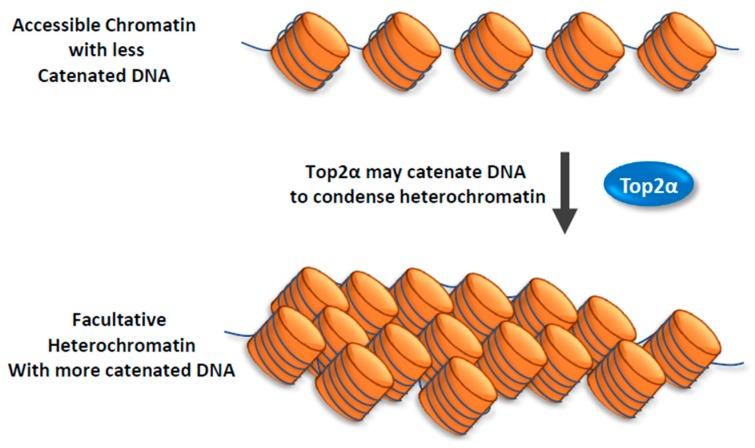
Top2α promotes the formation of facultative heterochromatin. A model illustrating how Top2α may facilitate the formation of facultative heterochromatin. Facultative heterochromatin may consist of DNA that is more catenated than that in the accessible chromatin. Top2α may catalyze DNA catenation during the formation of facultative heterochromatin from accessible chromatin. This model is based on the findings in a previous paper (Miller et al., Nat. Struct. Mol. Biol., 2017).

**Figure 3 genes-10-00884-f003:**
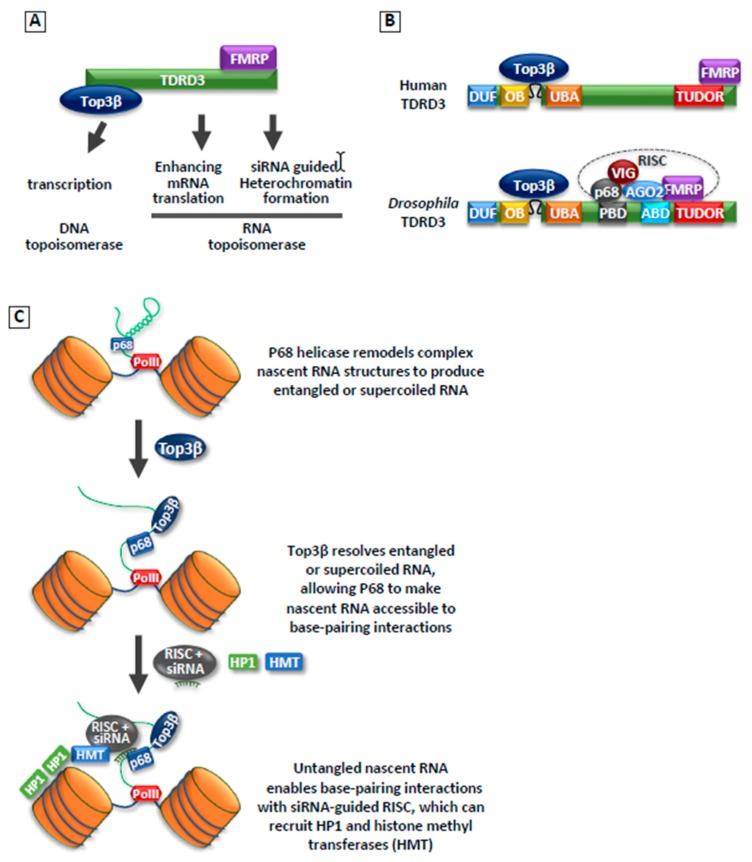
Top3β is a dual-activity topoisomerase that can act on both DNA and RNA, including RNAi-guided heterochromatin formation. (**A**) A cartoon illustrating that Top3β forms a complex with TDRD3, which interacts with FMRP. This complex can stimulate DNA transcription, mRNA translation, and siRNA-guided heterochromatin formation. (**B**) A cartoon illustrates human and *Drosophila* Top3β–TDRD3 complexes. In the latter species, Top3β–TDRD3 stably associates with the RNA-induced silencing complex (RISC) complex containing Argonaut 2 (AGO2), p68, FMRP, and VIG. It remains to be determined if mammalian Top3b–TDRD3 can also interact with RISC. (**C**) A model illustrates how Top3β may function during siRNA-guided heterochromatin formation. The model postulates that the unwinding of nascent transcript by p68 may generate entangled or supercoiled RNA that requires topoisomerase activity to resolve. Unwound RNAs by Top3β and p68 become more accessible for targeting by RISC to silence transposable elements (TE) as well as recruiting heterochromatin factors histone methyltransferase (HMT) and heterochromatin protein 1 (HP1).

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
