# Peer review of "Roles of Topoisomerases in Heterochromatin, Aging, and Diseases"

_genes, 2019, doi:10.3390/genes10110884_

Round 1

Reviewer 1 Report

The authors summarized the actual knowledge of heterochromatin changes in pathology and ageing, focusing on the role of topoisomerases in its formation and maintenance.

The text is clear, as well as figures, references are updated.

The paper provides a contribute in understanding the complex chromatin physiology and try to shed some light on the way employed by topoisomerases in solving topological chromatin problems to prevent cancer, ageing or neurodegenerative disorders.

Author Response

We thank reviewer 1 for his positive comments. We have gone through the manuscript and corrected spelling and grammar errors. 

Reviewer 2 Report

The review by Lee and Wang entitled “Roles of topoisomerases in heterochromatin, aging 2 and diseases” is a well written and easy to read review with a focus on the role of topoisomerases in maintaining heterochromatin architecture. This subject is not often highlighted and as such is very timely and warranted.

This manuscript describes the function of heterochromatin in cells. The authors clearly summarize the literature of the role of the various topoisomerases in maintaining heterochromatin and the consequences of topoisomerase defects on heterochromatin and human disease. They clearly correlate the data obtained in model systems to human cells work and biology. They also give their perspective on the current open questions in the field.

The only minor comment which needs to be addressed is the fact the they only highlight the difference between human and drosophila Top3b in the ledged of figure 3. This is in contrast to the other topoisomerases where they clearly explain the differences and similarities in the text. As such, the statement of “It remains to be determined if mammalian Top3b-TDRD3 can also interact with RISC” from figure 3 legend need to be further explained in the text. This is important for the context of what we know from model organism to human enzymes. Please, transfer the statement to the text and further discussed and highlighted differences and open questions.

Author Response

We thank Reviewer 2 for his/her positive comments on our manuscript. We have gone through our manuscript to correct grammatic and spelling errors. 

Regarding his minor comment, we have added a new paragraph to explain the difference between human and Drosophila Top3b-TDRD3 complexes as follows: 

"It should be mentioned that mammalian Top3b-TDRD3 complex differs from its Drosophila counterpart in that it lacks stable association with AGO2, as evidenced by the absence of AGO2 in Top3b or TDRD3 immunoprecipitation from human cancer cell lines (Xu et al. 2013). This result is consistent with the findings that mammalian TDRD3 lacks a recognizable AGO2-binding domain, which was detected in Drosophila TDRD3 (Lee et al. 2018). These data argue that the mechanism by which Top3b-TDRD3 complex works with RISC complex could be different in mammals versus flies. Notably, we and others have consistently observed that TDRD3 interacts with a RISC component, FMRP (Goulet et al. 2008; Linder et al. 2008; Stoll et al. 2013b; Xu et al. 2013). FMRP has been reported to interact with AGO2 (Li et al. 2014), and to regulate AGO2 binding to its target mRNAs in human cancer cell lines (Lee et al. 2010; Kenny et al. 2014; Skariah et al. 2017). Thus, one possibility is that mammalian Top3b-TDRD3 may interact indirectly with AGO2, and this interaction is mediated by FMRP. It remains to be determined if Top3b-TDRD3 functionally works with AGO2 in promoting heterochromatin formation and silencing of transposons in mammals."

We hope that this statement is clear enough for readers to understand the difference between human and Drosophila Top3b-TDRD3 on how they interact with RISC complex.